# Association between Grip Strength, Obesity, and Cardiometabolic Risk Factors among the Community-Dwelling Elderly Population in Taiwan

**DOI:** 10.3390/ijerph191811359

**Published:** 2022-09-09

**Authors:** Chun-Yung Chang, Nain-Feng Chu, Ming-Hsun Lin, Shu-Chuan Wang, Der-Min Wu, Ming-Kai Tsai, Chieh-Hua Lu

**Affiliations:** 1Department of Internal Medicine, Kaohsiung Armed Forces General Hospital, Kaohsiung City 802, Taiwan; 2Division of Occupational Medicine, Kaohsiung Veterans General Hospital, Kaohsiung City 813, Taiwan; 3School of Public Health, National Defense Medical Center, Taipei 114, Taiwan; 4Division of Endocrinology and Metabolism, Department of Internal Medicine, Tri-Service General Hospital, Taipei 114, Taiwan; 5Division of Performance Management, Kaohsiung Veterans General Hospital, Kaohsiung City 813, Taiwan

**Keywords:** grip strength, obesity, diabetes, cardiometabolic risk, elderly

## Abstract

The aim of this study was to evaluate the association between grip strength, obesity, and cardiometabolic risk factors among elderly individuals with different grip strength statuses and weight statuses in Taiwan. We conducted a series of community-based health surveys among the elderly population in Chiayi County, Taiwan from 2017 to 2019. This is a cross-sectionally designed health check-up program that was conducted by the local public health bureau. Anthropometric characteristics, handgrip strength, diabetes, and cardiometabolic risk profiles were measured using standard methods. This study recruited 3739 subjects (1600 males and 2139 females). The non-obese subjects had lower blood glucose (BG) levels compared to the obese subjects. The BG levels of non-obese and obese subjects were 102.7 ± 25.6 mg/dL vs. 109.1 ± 34.3 mg/dL for males; and 102.8 ± 30.1 mg/dL vs. 112.5 ± 40.3 mg/dL for females (both *p* < 0.001). The grip strength was negatively associated with BG in both sexes (β = −0.357, *p* < 0.001 for males and β = −0.385, *p* < 0.05 for females). The relationship between the grip strength and the risk of diabetes showed that for every 1 kg increase in the grip strength, there was a 4.1% and 4.5% decrease in the risk for developing diabetes for males and females, respectively (OR = 0.959, 95% CI = 0.940–0.979 for males and OR = 0.955, 95% CI = 0.932–0.978 for females). A higher handgrip strength is associated with a lower BG level and a lower risk for diabetes mellitus in the elderly Taiwanese subjects. Additional health promotion should focus on the obese and sarcopenic population to prevent cardiometabolic comorbidities in later life.

## 1. Introduction

Sarcopenia, which is defined as the age-related loss of skeletal muscle mass, was first described by Rosenberg in 1988 [1,2]. The diagnostic criteria of sarcopenia was defined by the European Working Group on Sarcopenia in Older People (EWGSOP) in 2010 by measuring the muscle mass, grip strength (GS), and gait speed [3]. In addition, the Asian Working Group for Sarcopenia (AWGS) in 2014 set a recommended cutoff for muscle quantity and quality in the Asian population [4].

In 2018, the revised EWGSOP2 used handgrip strength to screen for sarcopenia because it better predicted advanced outcomes [5]. The chair stand test was also a recommended screening tool, but is usually inaccessible when subjects are using wheelchairs or are unable to stand alone, which may limit the test’s ability to stratify the population for further analysis. Although the measurement of muscle mass remains a standard method in confirming sarcopenia, it is difficult and expensive to perform in a large-scale community health survey due to the need for standard instruments and trained technicians. Measuring the GS is a simple and inexpensive way to determine muscle strength and is also a good indicator for fragility [6,7]. Many studies have shown that muscle strength is a prognostic factor for all-cause death, cardiovascular death, and cardiovascular disease [6,8,9]. Moreover, surveys have demonstrated the relationship between muscle strength and cardiometabolic risk factors in the elderly [10,11]. The EWGSOP2 also suggested that a low GS and clinical suspicion is enough to start an assessment and intervention in a clinical practice setting. In our study, we measured GS and tried to evaluate its relationship with cardiometabolic risk factors.

Both the age-related loss of muscle and increased adipose tissues are important risk factors for cardiometabolic disorders in the elderly [12]. Obesity is associated with the occurrence of many chronic diseases such as hypertension, diabetes mellitus (DM), and dyslipidemia [13]. More recently, the concept of sarcopenic obesity had been established because of its shared association with metabolic disorders, morbidity, and mortality [12,14,15,16].

The past two decades have seen Taiwan transform into an aging society. Approximately 17.09% of Taiwan’s population in 2020 was aged 65 years and older. The municipality with the highest population of elderly individuals was Chiayi County, with 20.34% of its population being over the age of 65 [17]. More comprehensive health promotion programs for the elderly are urgently required in the Taiwan area. A similar study that evaluated the relationship between grip strength and cardiometabolic factors was conducted in middle-aged to elderly Taiwanese. However, the results demonstrated inconsistent findings among these CVD risks [18]. The aim of this study was to further evaluate the association between grip strength, obesity, and cardiometabolic risk factors among the community-dwelling elderly individuals with different weight statuses and handgrip strength statuses in Taiwan.

## 2. Materials and Methods

### 2.1. Study Population

We conducted a series of community-based health surveys among Chiayi County’s elderly individuals from 2017 to 2019. Those who were above 65 years old and had lived in Chiayi County for more than one year were invited to participate in the surveys. The individuals were invited during regular health check-ups or public health education programs by the local government. The inclusion criteria were individuals 65 to 85 years old. We excluded those who had any infectious diseases or acute disorders in the past three weeks preceding the start of the surveys.

### 2.2. Questionnaire

General demographic data including sex, age, residency, education level, occupation, and the need for a caregiver were collected using a standard questionnaire. Lifestyle patterns such as dietary habits, cigarette smoking, alcohol intake, and daily activity were also collected from the study population using the questionnaire.

### 2.3. Anthropometric Measurements

Anthropometric characteristics including body weight and body fat (BFAT) were measured using standard methods. Height was measured in meters using a digital stadiometer that recorded to the nearest 0.5 cm. Body weight was measured to an accuracy of 0.1 kg using a standard beam balance scale. During the measurement for the height, the subjects were barefoot and wore only light indoor clothing. The BFAT was obtained when subjects stood barefoot on the electrodes using a segmental body composition analyzer (TBF-410, Tanita Corp., Tokyo, Japan) and was expressed as percentages.

### 2.4. Grip Strength Measurement

The GS was measured using digital dynamometers (TKK5101). All subjects were in a seated position with fully extended elbows. After two to three minutes of rest, we measured the GS for either the dominant or non-dominant hand two times. Two values for the GS were recorded, and the mean value of the two recordings was used for analysis [19].

The subjects were divided into four subgroups according to their body weight classification and GS. Non-obese (OB−) subjects were defined as those with a BMI < 27, whereas obese (OB+) individuals were defined as those with a BMI ≥ 27. This threshold for obesity is defined based on the suggested physical status by the Health Promotion Administration, Ministry of Health and Welfare, Taiwan [20]. A normal grip strength (GS+) was defined as a GS ≥ 30 kg in the male subjects and a GS ≥ 20 kg in the female subjects, and a weak GS (GS−) was defined as a GS < 30 kg in the male subjects and a GS < 20 kg in the female subjects [3].

### 2.5. Blood Pressure Measurement

Blood pressure was obtained after the subjects had rested for five to ten minutes. In a seated position, the subjects’ arms were positioned at the same height as the heart and inserted into cuffs of appropriate sizes. Two measurements were recorded, and the mean value of the two recordings was used for data analysis.

### 2.6. Blood Specimen Collection

After 10–12 h of overnight fasting, 10 mL of venous blood was collected from the subjects using a venous container. The plasma and serum were separated from the blood within one hour and stored at −80 °C until analysis.

### 2.7. Plasma Glucose and Lipid Profile Measurement

The plasma glucose concentration of each subject was analyzed immediately after blood sampling and was determined using the glucose oxidase method via the Beckman Glucose Analyzer II (Beckman Instruments, Fullerton, CA, USA) [21]. We measured the total cholesterol (TC) level using an esterase oxidase method [22] and the triglyceride (TG) level using an enzymatic procedure [23] via a Hitachi 7150 auto-analyzer (Hitachi, Tokyo, Japan). High-density lipoprotein (HDLC) and low-density lipoprotein (LDLC) levels were measured using an enzymatic method [24].

### 2.8. Chronic Disease Status

The chronic disease status was acquired through a self-report history, including diabetes, hypertension, and dyslipidemia, under the assistance of a trained research technician. Additionally, we used the laboratory data cutoff point to define these disease statuses. For diabetes, it is defined as having a history of diabetes, currently being on anti-diabetes medications, or having a fasting blood glucose level ≥126 mg/dL. For hypertension, it is defined as having a history of hypertension, currently being on anti-hypertension medications, or having a systolic BP ≥140 mmHg or DBP ≥90 mmHg. For dyslipidemia, it is defined as having a history of dyslipidemia, currently being on lipid-lowering medications, or having a total cholesterol level ≥240 mg/dL, triglyceride level ≥200 mg/dL, or LDLC level ≥160 mg/dL. 

### 2.9. Approval of the IRB

All participants provided written informed consent and agreed to have their general demographic data, questionnaire, anthropometric data, and blood samples taken for this study. The Institutional Review Board of the Tri-Service General Hospital approved this study (number: TSGHIRB-1-108-05-073).

### 2.10. Statistical Methods

We used SPSS ver. 22 (IBM Corp., Armonk, NY, USA) to conduct all statistical analyses. Continuous variables, such as anthropometric measures, grip strength, and the cardiometabolic risk profile, were presented as sample mean and SD. The Mann–Whitney U test was used to compare the differences between the groups. The Kruskal–Wallis H test was used to compare more than three groups. The categorical variables were described as numbers and percentages. We used a frequency histogram to check whether continuous random variables had a skewed or non-skewed distribution. For TG, we used a log transformation to reduce the skewness of a measurement variable. The chi-square test was used to compare the differences among two or more groups. We used multivariant regression analyses and logistic regression analyses for further statistical inference. A two-tailed *p*-value less than 0.05 was considered statistically significant.

## 3. Results

The demographics, anthropometrics, and cardiometabolic risk profiles among the study population were distributed into quartiles with sex and age specifications and are presented in Table 1. A total of 3739 subjects (1600 males and 2139 females) were recruited. The mean GS was 32.8 ± 7.2 kg for males and 21.6 ± 4.8 kg for females. The subjects were divided into four subgroups based on their specific GS (Q1–Q4, lowest to highest). In both sexes, the group with the highest GS also had the highest BMI and BFAT (*p* < 0.001).

Table 2 shows the distribution of the cardiometabolic risk profile among the study population according to their weight classification and GS status with sex specification. The non-obese (OB−) groups had lower systolic blood pressure (SBP), TG levels, and blood glucose (BG) levels, and higher HDLC levels compared to the obese (OB+) groups. Among the female subjects, the OB− groups had a lower diastolic blood pressure (DBP) than the OB+ groups. Among the OB− groups, the BG levels and SBP were lower in the GS+ subgroups than the GS- subgroups. The mean BG level for the male subjects was 102.7 ± 25.6 mg/dL and 104.3 ± 33.3 mg/dL in the OB−/GS+ and the OB−/GS− subgroups, respectively (*p* < 0.001). The mean BG level for the female subjects was 102.8 ±30.1 mg/dL and 103.3 ± 31.4 mg/dL in the OB−/GS+ and the OB−/GS− subgroups, respectively (*p* < 0.001). The mean SBP for the male subjects was 137.2 ± 17.8 mmHg and 137.7 ± 19.5 mmHg in the OB−/GS+ and the OB−/GS− subgroups, respectively (*p* < 0.001). The mean SBP for the female subjects was 138.6 ± 17.5 mmHg and 138.7 ± 18.5 mmHg in the OB−/GS+ and the OB−/GS− subgroups, respectively (*p* < 0.001).

The multivariable regression analyses for the GS in relation to the cardiometabolic risk profiles before and after adjusting for the potential confounders with sex specifications is shown in Table 3. In both sexes, the GS was negatively associated with BG (β = −0.357, *p* < 0.001 for males and β = −0.385, *p* < 0.05 for females) and positively associated with TC (β = 0.658, *p* < 0.001 for males and β = 0.792, *p* < 0.001 for females) and LDLC (β = 0.489, *p* < 0.001 for males and β = 0.604, *p* < 0.001 for females). In the male subjects, the GS was positively associated with DBP (β = 0.167, *p* = 0.005) and HDLC (β = 0.180, *p* = 0.001). In the female subjects, the GS was positively associated with SBP (β = 0.305, *p* = 0.001) only.

Table 4 demonstrates the GS in relation to the risk for cardiometabolic disease. For every 1 kg increase in the GS, there was a decrease of 4.1% and 4.5% in the risk for DM in the male and female subjects, respectively (OR = 0.959, 95% CI = 0.940–0.979 for the male subjects and OR = 0.955, 95% CI = 0.932–0.978 for the female subjects). However, a higher GS seems to be associated with an increased risk for dyslipidemia in the female subjects (OR = 1.029, 95% CI = 1.006–1.053).

## 4. Discussion

In the present study, we found that the non-obese subjects had lower blood pressure, triglyceride and BG levels compared with the obese subjects. The subjects who had a higher GS had lower BG levels than the subjects with a lower GS. Our results revealed that the GS may be a protective factor against DM and cardiovascular disease among the community-dwelling elderly population in Taiwan.

In distributing the study population into quartiles based on grip strength, there was no significant difference in the blood glucose level and blood pressure among the different quartile subgroups. It was possible that the increased BMI and body fat percentage negated the effect of GS on the BG level since obesity itself plays an important role in the development of hyperglycemia, insulin resistance, and metabolic syndrome [25,26]. This was also observed in the subjects that were distributed into four groups by GS and BW classification. The levels of the metabolic syndrome indicators (BG, SBP, TG, and HDLC) in the non-obese subjects were ideal compared to those of the obese subjects.

The associations between the GS, lipid profiles, and risk for dyslipidemia were not consistent in our study. A higher GS was associated with higher levels of TC, HDLC, and LDLC in males and higher TC and LDLC levels in females, and the risk for dyslipidemia was only increased in the female subjects. This finding was similar to the result of the Korea National Health and Nutrition Examination Survey (KNHANES), which GS was shown to have a positive association with an unfavorable lipid biomarker in the univariate linear regression [27]. Additionally, a Switch study showed the GS to only have a moderate association with the cardiovascular risk markers. These results suggest that the GS may have a complex association with the lipid profile since a subject with extremely low GS values might be associated with a “deleterious low” lipid profile [28] due to poor nutrition status. Further longitudinal studies including detailed nutrition status and medical history may be considered.

In our study, the GS was positively associated with SBP in males and DBP in females. However, the earlier systemic review and meta-analysis demonstrated sarcopenia to be associated with hypertension [29]. Our next study will focus on the direct relationship between GS, detailed body composition, and BP to investigate the conflicting results. 

After adjusting for age and body composition, the GS had an inverse association with the BG level in our study. It was similar to previous results that demonstrated the favorable correlation of GS to the fasting glucose and glycohemoglobin (HbA1c) levels in women [18]. Our results were also similar to the findings of the KNHANES, which demonstrated the inverse association between the relative GS and fasting glucose/HbA1c levels [7]. In another study among the Asian population, Liang et al. reported the inverse dose–response association between the GS and fasting BG level [30]. In our study, the GS is a protective factor against the development of DM for both sexes. This result was consistent with several studies that have evaluated the relationship between the GS and cardiometabolic diseases. The results of the Healthy Life in an Urban Setting (HELIUS) Study, which included six ethnic groups, also revealed the inverse correlation between the GS and DM prevalence [31]. Similarly, the Helsinki Birth Cohort Study (HBCS) found that the GS was lower in those with known and newly diagnosed DM compared to those with normal BG levels [32]. Further studies conducted by the KNHANES also showed that the GS was negatively associated with the development of type 2 DM and insulin resistance [27,33].

Although there were several studies that have demonstrated the association between the GS and DM, their causal relationship remains unclear. Long-term exercise and training that might be related to higher muscle strength attenuates lipid-induced insulin resistance [34]. In contrast, a lower muscle strength might be caused by inflammation, which is also an important factor caused by insulin resistance [35]. However, further mechanistic studies and trials are needed to determine the causal relationship between the GS and BG levels.

### Limitations

There are still some limitations in our study. First, the cross-sectional design without follow-ups limits its ability to evaluate the causal relationships between GS, diabetes, and cardiometabolic risk profiles. Second, we did not measure muscle mass in the study. Although GS was a convenient and effective way to detect and screen for sarcopenia among the elderly, the muscle mass and GS should have been measured at the same time to properly assess for the presence of sarcopenia [4,5]. Third, we did not include all possible cardiometabolic risk factors in this study. For example, serum uric acid (SUA) is also an important risk factor for cardiometabolic disease [36,37]. However, we did not collect information about SUA in this study, which may influence the power of our final model. Finally, the status of chronic diseases, such as diabetes, hypertension, and dyslipidemia, were acquired by self-report information in the questionnaire. Thus, informational bias regarding the history of chronic diseases in the subjects could not be excluded. 

## 5. Conclusions

In general, the obese subjects had higher BG levels than the non-obese subjects. Furthermore, after adjusting for age and body composition, a higher GS was associated with a lower BG level and lower risk for DM in the community-dwelling elderly Taiwanese population. Additional health promotion studies to increase muscle mass along with early rehabilitation programs should focus on the obese and sarcopenic population to prevent cardiometabolic comorbidities in later life.

## Figures and Tables

**Table 1 ijerph-19-11359-t001:** Distributions of anthropometric variables and cardiometabolic risk factors among study population by quartile grip strength subgroups with sex specification (*n* = 3739) (mean ± SD).

	Grip Strength (kg)								
	Q1 (Lowest)		Q2		Q3		Q4 (Highest)		*p*-Value ^†^
**Male**	<28		28–32.85		32.85–37.4		>37.4		
(*n* = 1600)	(*n* = 399)		(*n* = 401)		(*n* = 397)		(*n* = 403)		
Height ^$^ (cm)	159.0	±5.7	161.2	±5.5	163.4	±5.2	165.7	±5.6	<0.001
Weight (kg)	61.7	±9.8	63.9	±9.4	66.8	±9.0	70.8	±9.8	<0.001
BMI (kg/m^2^)	24.3	±3.5	24.6	±3.4	25.0	±3.3	25.8	±3.3	<0.001
BFAT (%)	22.3	±6.8	22.1	±6.8	22.5	±5.9	24.0	±5.8	<0.001
Systolic BP (mmHg)	138.4	±19.3	139.5	±18.6	137.7	±17.7	139.2	±17.0	0.489
Diastolic BP (mmHg)	78.7	±11.4	81.1	±10.5	82.3	±11.0	84.6	±10.5	<0.001
BG (mg/dL)	105.9	±33.3	104.7	±30.6	106.4	±30.7	101.7	±23.9	0.119
TC (mg/dL)	177.6	±36.9	179.6	±33.7	184.6	±35.1	186.4	±34.5	0.001
TG (mg/dL)	115.5	±66.6	116.3	±82.3	120.0	±75.5	125.2	±73.1	0.033
LDLC (mg/dL)	97.7	±30.7	98.7	±28.2	103.0	±29.2	106.0	±30.7	<0.001
HDLC (mg/dL)	50.9	±13.7	52.5	±14.1	52.3	±13.3	51.5	±13.3	0.315
**Female**	<18.2		18.2–21.7		21.7–24.7		>24.7		
(*n* = 2139)	(*n* = 531)		(*n* = 544)		(*n* = 544)		(*n* = 520)		
Height (cm)	148.3	±5.8	149.9	±5.4	151.7	±5.1	153.6	±5.2	<0.001
Weight (kg)	54.2	±9.3	55.7	±9.0	57.8	±8.9	61.0	±9.1	<0.001
BMI (kg/m^2^)	24.6	±4.0	24.8	±3.9	25.1	±3.7	25.9	±3.8	<0.001
BFAT (%)	31.1	±7.6	31.8	±7.3	32.6	±7.4	34.1	±6.8	<0.001
Systolic BP (mmHg)	139.9	±18.9	140.8	±17.8	139.4	±17.8	140.9	±17.7	0.414
Diastolic BP (mmHg)	77.4	±10.5	78.8	±10.5	78.4	±10.0	80.2	±9.8	<0.001
BG (mg/dL)	105.7	±32.7	105.3	±36.6	106.4	±35.2	104.5	±30.8	0.850
TC (mg/dL)	194.5	±38.7	198.6	±36.3	199.2	±37.0	205.0	±38.4	<0.001
TG (mg/dL)	124.3	±72.7	122.9	±63.0	123.6	±70.1	127.8	±74.9	0.763
LDLC (mg/dL)	104.6	±31.7	108.8	±30.1	107.3	±30.0	112.9	±31.1	<0.001
HDLC (mg/dL)	59.4	±15.0	59.7	±14.7	60.5	±14.3	60.1	±13.9	0.627

^$^ Abbreviations: Height, body height; Weight, body weight; BMI, body mass index; BFAT, body fat; Systolic BP, systolic blood pressure; Diastolic BP, diastolic blood pressure; BG, blood glucose; TC, total cholesterol; TG, triglyceride; LDLC, low-density lipoprotein cholesterol; HDLC, high-density lipoprotein cholesterol. ^†^ ANOVA F test was used to compare the anthropometric variables and cardiometabolic risk factors among these four subgroups with sex specifications; TG used log transformation to test.

**Table 2 ijerph-19-11359-t002:** Distributions of cardiometabolic risk factors among study population by different weight status and grip strength status with sex specification (*n* = 3739) (mean ± SD).

	Obesity Status and Grip Strength Status								
	OB(−), GS(−)		OB(−), GS(+)		OB(+), GS(+)		OB(+), GS(−)		*p*-Value ^†^
**Male** (*n* = 1600)	(*n* = 431)		(*n* = 772)		(*n* = 289)		(*n* = 108)		
Systolic BP ^$^ (mmHg)	137.7	±19.5	137.2	±17.8	142.6	±16.7	143.3	±17.2	<0.001
Diastolic BP (mmHg)	78.7	±11.2	82.3	±10.7	84.7	±10.4	81.1	±11.7	<0.001
BG (mg/dL)	104.3	±33.3	102.7	±25.6	109.1	±34.3	109.1	±29.1	0.007
TC (mg/dL)	179.3	±36.4	184.3	±34.3	182.3	±36.3	176.1	±33.0	0.029
TG (mg/dL)	108.4	±62.7	114.3	±75.1	144.1	±86.5	131.4	±65.3	<0.001
LDLC (mg/dL)	99.1	±29.8	102.9	±29.5	102.6	±31.0	96.3	±28.7	0.046
HDLC (mg/dL)	53.0	±14.1	53.3	±13.7	47.7	±11.9	47.1	±11.7	<0.001
**Female** (*n* = 2139)	(*n* = 573)		(*n* = 969)		(*n* = 408)		(*n* = 189)		
Systolic BP (mmHg)	138.7	±18.5	138.6	±17.5	144.5	±18.0	143.8	±17.5	<0.001
Diastolic BP (mmHg)	76.7	±10.2	78.4	±10.0	81.1	±10.0	80.9	±11.1	<0.001
BG (mg/dL)	103.3	±31.4	102.8	±30.1	112.5	±40.3	110.4	±41.5	<0.001
TC (mg/dL)	194.4	±37.1	202.2	±37.9	199.7	±35.7	198.4	±42.1	0.001
TG (mg/dL)	120.0	±66.1	118.0	±67.6	139.1	±70.7	141.1	±86.2	<0.001
LDLC (mg/dL)	104.7	±30.4	110.3	±31.1	109.6	±28.6	107.0	±34.6	0.005
HDLC (mg/dL)	59.7	±15.1	61.6	±14.9	57.2	±11.9	58.5	±14.3	<0.001

^$^ Abbreviations: Systolic BP, systolic blood pressure; Diastolic BP, diastolic blood pressure; BG, blood glucose; TC, total cholesterol; TG, triglyceride; LDLC, low-density lipoprotein cholesterol; HDLC, high-density lipoprotein cholesterol. ^†^ ANOVA F test was used to compare the cardiometabolic risk factors among these four subgroups with sex specifications; TG used log transformation to test. OB(−): non-obese, BMI < 27; OB(+): obese, BMI ≥ 27. GS(−): weak grip strength, GS < 30 for males and GS < 20 for females. GS(+): normal grip strength, GS ≥ 30 for males and GS ≥ 20 for females.

**Table 3 ijerph-19-11359-t003:** Multivariate regression analyses of grip strength on cardiometabolic risk factors in different models with sex specification.

Dependent Variables	Model I ^†^			Model II ^‡^		
	β	se β	*p*-Value	β	se β	*p*-Value
**Male (*n* = 1600)**						
Systolic BP ^$^ (mmHg)	0.200	0.071	0.005	0.118	0.077	0.123
Diastolic BP (mmHg)	0.254	0.043	<0.001	0.167	0.127	0.005
BG (mg/dL)	−0.228	0.118	0.052	−0.357	0.074	<0.001
TC (mg/dL)	0.514	0.138	<0.001	0.658	0.151	<0.001
TG (mg/dL)	0.234	0.293	0.424	−0.081	0.313	0.797
LDLC (mg/dL)	0.442	0.117	<0.001	0.489	0.128	<0.001
HDLC (mg/dL)	0.043	0.054	0.427	0.180	0.056	0.001
**Female (*n* = 2139)**						
Systolic BP (mmHg)	0.395	0.085	<0.001	0.305	0.088	0.001
Diastolic BP (mmHg)	0.152	0.049	0.002	0.088	0.051	0.087
BG (mg/dL)	−0.143	0.164	0.382	−0.385	0.170	0.024
TC (mg/dL)	0.569	0.181	0.002	0.792	0.190	<0.001
TG (mg/dL)	0.252	0.339	0.458	−0.172	0.350	0.623
LDLC (mg/dL)	0.458	0.148	0.002	0.604	0.156	<0.001
HDLC (mg/dL)	−0.030	0.070	0.672	0.122	0.072	0.089

^$^ Abbreviations: β, regression coefficient; se, standard error; Systolic BP, systolic blood pressure; Diastolic BP, diastolic blood pressure; BG, blood glucose; TC, total cholesterol; TG, triglyceride; LDLC, low-density lipoprotein cholesterol; HDLC, high-density lipoprotein cholesterol. ^†^ Model I: adjusting for age. ^‡^ Model II: further adjusting for body height, body weight, body fat, smoking, and alcohol drinking.

**Table 4 ijerph-19-11359-t004:** Multivariate logistic regression analyses of grip strength on cardiometabolic diseases in different models with sex specification.

Dependent Variables	Model I ^†^		Model II ^‡^	
	OR ^$^	95% CI	OR	95% CI
**Male**				
Hypertension	1.021	1.004–1.038	1.011	0.992–1.030
Diabetes mellitus	0.971	0.954–0.989	0.959	0.940–0.979
Dyslipidemia	1.011	0.995–1.027	1.006	0.988–1.023
**Female**				
Hypertension	1.016	0.995–1.037	0.997	0.975–1.020
Diabetes mellitus	0.972	0.951–0.995	0.955	0.932–0.978
Dyslipidemia	1.036	1.014–1.058	1.029	1.006–1.053

^$^ Abbreviations: OR, odds ratio; CI, confidence interval. ^†^ Model I: adjusting for age. ^‡^ Model II: further adjusting for body height, body weight, body fat, smoking, and alcohol drinking.

## Data Availability

Data are available from the corresponding author.

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
