# Peer review of "Association between Grip Strength, Obesity, and Cardiometabolic Risk Factors among the Community-Dwelling Elderly Population in Taiwan"

_ijerph, 2022, doi:10.3390/ijerph191811359_

Round 1
Reviewer 1 Report
The authors performed a cross-sectional study in >3700 taiwanese people aged 65-85 years and correlated BMI, grip strength, cholesterol levels and blood sugar with self-assessed presence of cardiometabolic risk factors. They found higher handgrip strength to be associated with lower blood glucose level and lower risk for diabetes mellitus. Furthermore, they found non-obese participants to have lower blood glucose levels, lipid levels and blood pressure.
This is a very interesting cross-sectional study with well-performed methodology and statistics. The manuscriopt is written in good English. However, as several different associations were compared, the manuscript is quite difficult to read without a clear conclusion. I therefore recommend that the authors should emphasize on one or two outcomes, for example obesity and handgrip strength, only.
Minor comments:
· In the abstract, full results of the given main outcome should be presented, please avoid „for example“
· The differentiation between female and male participants makes analysis quite complicated. As results are different in both genders, the interpretation of the results is even more complicated.
· The authors should write in the Introduction WHY they performed this study. Is there currently limited data about this topic available?
· Why did the authors choose 27 as threshold for obesity?
· Table 1 is quite difficult to interpret and has no clear result. Maybe the authors could concentrate on results from the following tables instead.
· Table 3: Results from forced parameters should be included (body height, body weight, waist to hip ratio, body fat, smoking, and alcohol drinking)
· Tables: it is difficult to read results as all parameters are abbreviated.
· The authors should clarify in the abstract that this study is cross-sectional without follow up. Furthermore, the authors should take care when interpreting the results, as only association and no causation can be derived from this study. This may also be noted in the Limitations section.
· I recommend to add a new section „Limitations“ and include those limitations written at lines 210-220. This section may suit best immediately before the Conclusions section.
· More information about recruitment (how) would be favourable.
· Line 231: what does the following sentence (especially „while“) mean: While the risk for dyslipidemia was only increased in the female subjects.
Author Response
Response to Reviewer 1 Comments
The authors performed a cross-sectional study in >3700 Taiwanese people aged 65-85 years and correlated BMI, grip strength, cholesterol levels and blood sugar with self-assessed presence of cardiometabolic risk factors. They found higher handgrip strength to be associated with lower blood glucose level and lower risk for diabetes mellitus. Furthermore, they found non-obese participants to have lower blood glucose levels, lipid levels and blood pressure.
This is a very interesting cross-sectional study with well-performed methodology and statistics. The manuscript is written in good English. However, as several different associations were compared, the manuscript is quite difficult to read without a clear conclusion. I therefore recommend that the authors should emphasize on one or two outcomes, for example obesity and handgrip strength, only.
Thank you for your valuable comments. We did our best to revise our manuscript and we focus on the association of obesity and grip strength on diabetes and CVD risk factors.
Minor comments:
- In the abstract, full results of the given main outcome should be presented, please avoid “for example”
Thank you for your valuable comments. We deleted the “for example” and replaced it with the full results of the outcome in the updated manuscript. (Relationship between blood glucose, obesity, and grip strength). (As page 2, line 40-41).
- The differentiation between female and male participants makes analysis quite complicated. As results are different in both genders, the interpretation of the results is even more complicated.
Thank you for your comment. Because sex is a major determinant of grip strength and physical status (such as anthropometrics and CVD risk factors…), we conduct further analyses with sex-specification to explore the difference between sexes. In this study, the relationship between grip strength, blood glucose, and diabetes are similar.
- The authors should write in the Introduction WHY they performed this study. Is there currently limited data about this topic available?
Thank you for your comment. Although there were some studies that examine the relationship between grip strength and CVD risks, but the results were inconsistent. In our study, we focused on community-dwell elderly and tried to find the relationship between cardiometabolic risk factors and grip strength. This model may help us improve our health promotion programs in the future. We added and revised this in our updated manuscript. (As page 4, line 95-99).
- Why did the authors choose 27 as threshold for obesity?
Thank you for your precious comment. Since this study is a localized community study, we choose the threshold which is suitable for Taiwanese population. BMI ≥ 27 as threshold for obesity was set according to the Evidence-based Guideline on Adult Obesity Prevention and Management, suggested by the Health Promotion Administration, Ministry of Health and Welfare in Taiwan, and Taiwan Medical Association for the Study of Obesity. (As page 5, line 139-141).
- Table 1 is quite difficult to interpret and has no clear result. Maybe the authors could concentrate on results from the following tables instead.
We revised this Table in our updated manuscript. Thank you for your comment.
- Table 3: Results from forced parameters should be included (body height, body weight, waist to hip ratio, body fat, smoking, and alcohol drinking)
We added and revised these variables into the model in our updated manuscript. Thank you for your comment.
- Tables: it is difficult to read results as all parameters are abbreviated.
We revised these variables with full names in our updated manuscript. Thank you for your comment.
- The authors should clarify in the abstract that this study is cross-sectional without follow up. Furthermore, the authors should take care when interpreting the results, as only association and no causation can be derived from this study. This may also be noted in the Limitations section.
Thank you for your precious comment. We revised our statement in the abstract and in the updated manuscript. The limitations were also described in a new section as the next comment. (As page 2, line 35-36 and page 10, line 302-304).
- I recommend to add a new section “Limitations” and include those limitations written at lines 210-220. This section may suit best immediately before the Conclusions section.
We added “Limitations” section and noted this study as a cross-sectional design with several limitations in our updated manuscript. Thank you for your comment. (As page 10, line 305-314).
- More information about recruitment (how) would be favourable.
Thank you for your valuable comments. We added and revised this in our updated manuscript. Thanks again. (As page 4, line 110-112).
- Line 231: what does the following sentence (especially „while“) mean: While the risk for dyslipidemia was only increased in the female subjects.
Thank you for the precious comment. We try to state that the risks for dyslipidemia only increased in the female subjects, but not in male subjects. We adjusted our statement to avoid confusion in the updated manuscript.

Reviewer 2 Report
Dear Editor,
I carefully read the manuscript by Chang et al.
My comments and suggestions are the following:
- Line 74: The authors wrote to have collected information regarding "gender". Are they sure they investigated gender and not participants' sex?
- Lines 74-80: Were the used questionnaires validated? The authors should include in the manuscript references supporting the use of the questionnaires in the study.
- Lines 119-120: "Total cholesterol" should be more appropriately abbreviated as "TC".
- Line 129: Statistical methods should be described more accurately. For example, the authors should declare how the normal distribution of the variables was assessed.
- The authors should include in the "Methods" how dyslipidemia, hypertension and diabetes mellitus were diagnosed (i.e. which guidelines the authors referred to). Appropriate references should be added to the manuscript.
- The limitations of the study should be more deeply discussed.
- Is information regarding serum uric acid (SUA) in population available? The authors should also comment on this topic. As a matter of fact, SUA has been associated to an amount of cardiometabolic risk factors in general population (doi: 10.1038/s41598-018-29955-w and doi: 10.1080/07853890.2016.1222451)
Author Response
Response to Reviewer 2 Comments
My comments and suggestions are the following:
- Line 74: The authors wrote to have collected information regarding "gender". Are they sure they investigated gender and not participants' sex?
Thank you for the precious comment. We try to explore the difference between genders. However, we only collected the information regarding "sex" rather than “gender”. We had revised this statement to avoid confusion in the updated manuscript.
- Lines 74-80: Were the used questionnaires validated? The authors should include in the manuscript references supporting the use of the questionnaires in the study.
Thank you for your precious comment. The questionnaires used collected demographic data including sex, age, residency, education level, occupation, lifestyle, and the need for a caregiver using standard questionnaires. The anthropometric and cardiometabolic risk parameters were measured using standard methods, with supporting data in the references. The diseases statuses were defined using either questionnaire information or laboratory data. We added and revised this in the method section in the updated manuscript.
- Lines 119-120: "Total cholesterol" should be more appropriately abbreviated as "TC".
We revised this abbreviation as “TC” in the updated manuscript. Thank you for the comment.
- Line 129: Statistical methods should be described more accurately. For example, the authors should declare how the normal distribution of the variables was assessed.
Thank you for your comments. We used a frequency histogram to check whether the distribution for continuous random variables were skewed or non-skewed. For TG, we used log-transformed to reduce the skewness of a measurement variable. We added and revised this in the updated manuscript. (As page 6, line 189-191).
- The authors should include in the "Methods" how dyslipidemia, hypertension and diabetes mellitus were diagnosed (i.e. which guidelines the authors referred to). Appropriate references should be added to the manuscript.
Thank you for your comments. To explore and control data, we used a structure questionnaire to collect participants’ history and medication of chronic diseases status (i.e. dyslipidemia, hypertension and diabetes mellitus mills) with the assistance of a trained research technician. We also used the laboratory data cut-off point to define these diseases’ status. We added and revised the method section to make this point clearer. Thanks again. (As page 6, line 166-175).
- The limitations of the study should be more deeply discussed.
Thank you for the precious comment. We added and revised the new “Limitations” section to discuss the limitations separately in our updated manuscript. (As page 10, line 305-314).
- Is information regarding serum uric acid (SUA) in population available? The authors should also comment on this topic. As a matter of fact, SUA has been associated to an amount of cardiometabolic risk factors in general population (doi: 10.1038/s41598-018-29955-w and doi: 10.1080/07853890.2016.1222451)
Thank you for your comments on this issue. SUA became an important CVD risk factor in recently study findings. However, we did not collect the information about SUA in this study. We will add this as limitations of our study and will try to collect further information in the future study. Thanks again. (As page 10, line 307-311).

Round 2
Reviewer 1 Report
All comments were addressed appropriately.